# Tracing Worldwide Turkey Genetic Diversity Using D-loop Sequence Mitochondrial DNA Analysis

**DOI:** 10.3390/ani9110897

**Published:** 2019-11-01

**Authors:** Amado Manuel Canales Vergara, Vincenzo Landi, Juan Vicente Delgado Bermejo, Amparo Martínez, Patricia Cervantes Acosta, Águeda Pons Barro, Daniele Bigi, Phillip Sponenberg, Mostafa Helal, Mohammad Hossein Banabazi, María Esperanza Camacho Vallejo

**Affiliations:** 1Department of Genetics, Faculty of Veterinary Sciences, University of Córdoba, ceiA3. 14071 Cordoba, Spain; landivincenzo@yahoo.it (V.L.); juanviagr218@gmail.com (J.V.D.B.); amparomartinezuco@gmail.com (A.M.); 2Facultad de Medicina Veterinaria y zootecnia, Universidad Veracruzana, Veracruz 91710, Mexico; biomoluv@gmail.com; 3Serveis de Millora Agrària (SEMILLA), 07009 Palma de Mallorca, Spain; apons@semilla-caib.es; 4Department of Agricultural and Food Sciences, Division of Animal Sciences, University of Bologna, Viale G. Fanin 46, 40127 Bologna, Italy; daniele.bigi@unibo.it; 5Virginia-Maryland College of Veterinary Medicine, Blacksburg, VA 24060, USA; dpsponen@vt.edu; 6Faculty of Agriculture, Cairo University, Giza 12613, Egypt; mostafa.helal@agr.cu.edu.eg; 7Department of Biotechnology, Animal Science Research Institute, Karaj 3146618361, Iran; banabazi@ut.ac.ir; 8Instituto de Investigación y Formación Agraria y Pesquera (IFAPA), Alameda del Obispo, 14004 Córdoba, Spain; mariae.camacho@juntadeandalucia.es

**Keywords:** *Meleagris gallopavo*, mtDNA, phylogenetic relationships, genetic diversity, populations

## Abstract

**Simple Summary:**

The development of new production lines of turkeys has relegated native breeds to a second position. This has increased the need for new research to ensure the conservation of local turkey breeds and the maintenance of biodiversity. The objective of the present study was to identify turkey populations, their origins, and maternal lines through mitochondrial DNA analysis. For this study, mitochondrial DNA samples from 93 turkeys *(Meleagris gallopavo)* were used. The animals belonged to populations in Brazil, Mexico, Spain (Andalusia and Majorca) Italy, Iran, Egypt, and the United States. The haplogroup network that formed suggested that turkey domestic populations group into a single haplotype. However, genetic differences within the haplogroup were found. The present study may provide a better approach for the implementation of conservation strategies for domestic turkey populations.

**Abstract:**

According to recent archeological evidence, turkey *(Meleagris gallopavo gallopavo)* domestication may have occurred in Mexico around 2000 years ago. However, little is known about the phylogenetic and genealogical background underlying domestic turkey populations. This study aimed to further understand the domestication process and identify inter- or intraspecific connections between turkey populations to determine their origins, trace their global expansion, and define the species’ genetic value. Ninety-three domestic turkeys (local breeds) were sampled from populations in Brazil, Mexico, USA, Spain, Italy, Iran, and Egypt. Publicly available sequences from previous studies were also included. Standard mitochondrial DNA, genetic diversity, and haplotype network analyses were performed. Seventy-six polymorphic sites were identified. Turkeys from Mexico showed the greatest number of polymorphic sites (40), while turkeys from Italy and Brazil reported only one site each. Nucleotide diversity was also highest in Mexico and the USA (π = 0.0175 and 0.0102, respectively) and lowest in Brazil and Italy. Of the six major haplogroups defined, the Mexican and USA populations appeared to have remained more stable and diverse than the other populations. This may be due to conservative husbandry policies in the rural areas of other populations, which have prevented the introduction of commercial turkey lines.

## 1. Introduction

Despite the global economic importance of turkeys (*Meleagris gallopavo)*, scientific efforts to understand the genetic diversity of the species have been infrequent. *M. gallopavo* is presumably the only important North-American domestic species of turkey [1]. Geographical and morphological criteria have been used to identify and describe six subspecies in North American territories [2], namely *Meleagris gallopavo gallopavo* (domestic Mexican subspecies), *Meleagris gallopavo silvestris* (wild), *Meleagris gallopavo mexicana* (Gould), *Meleagris gallopavo intermedia* (Rio Grande), *Meleagris gallopavo osceola* (Florida), and *Meleagris gallopavo merriami* (Merriam) [3]. The Mexican subspecies is generally accepted as the ancestral origin and the only important domestic population from which the other subspecies may have descended. This is due to its particular ability to adapt to tropical and humid climates in North America [1,4], which provided the basis for the successful introduction of the species to southeastern Canada and to the eastern and southern regions of the USA after its reintroduction by British European Pilgrims in 1620 [1].

Mitochondrial DNA (mtDNA)-based molecular studies have been performed on domestic turkey populations in order to trace the history of the extinct wild subspecies, *M*. *g*. *gallopavo* [5]. Historical registries have suggested that different prehispanic Mexican groups, like the Teotihuacans, began the domestication of the turkey for the first time between 200 and 700 BC [4]. Considering this context and that the fact that, according to the hypothesis suggested by Flannery [6], Leopold [7], and Schorger [8], an exact time for domestication cannot be clearly identified, the highlands of Michoacan, Mexico have been proposed as the most likely original focus for the species. This is supported by the most likely archaeological hypotheses, based on findings within the Tehuacan Valley (Puebla). Bones dating from approximately 700 AD have also been identified in Guatemala, which suggests the southern expansion of the species by that time [9].

The definition of zoogenetic resources and the characterization of local populations are challenging but necessary when aiming to develop biodiversity conservation policies [10]. In this context, the use of mtDNA sequences in phylogeographical analyses has been extensively explored and has offered a highly sensitive method to analyze the origin and evolutionary processes of different species [1,11].

All the main current domestic turkey varieties descend from the native turkey of central Mexico, which was subsequently imported into Europe during Spanish colonization and was later distributed throughout the extensive Spanish territories. However, it was not until the seventeenth century that turkeys spread across Europe, South America, Africa, and many countries of the Middle East as an alternative to pork. The relationships among wild turkeys have been extensively evaluated, primarily for conservation purposes, using both morphological and mitochondrial molecular tools, such as amplified fragment length polymorphism (AFLP), DNA markers, and mitochondrial control region analysis by using heterologous primers from the chicken. Mock et al. [2] evaluated the genetic variation among wild turkeys widely distributed in the USA. In their study, molecular analyses revealed relationships among turkeys from distinct geographical regions that were also consistent with earlier morphological definitions of turkey varieties [2].

Differences among turkey varieties have also been investigated at the molecular level using diverse nuclear DNA marker systems [12,13], including microsatellites and single nucleotide polymorphisms (SNPs). mtDNA assessment can explain and provide additional support to the evaluation of distinctions between local domestic breeds and other heritage and commercial turkey breeds (industrial meat production), based on their relatedness [14]. Presently, several studies of turkey characterization are ongoing. For instance, African populations are being characterized based on their external characteristics [15]. However, studies on the genetic background of domestic populations of this species are scarce, with only a few molecular genetic studies reported on domestic turkeys from rural communities in Spain, Mexico, USA, Iran, Italy, Egypt, and Brazil.

The present study aimed to analyze mitochondrial diversity in several domestic breeds of turkey from seven different countries located on four different continents (America, Europe, Asia, and Africa). Original regions and derivate populations were included in the study. Firstly, we analyzed sequences of the mtDNA D-loop region from domestic, commercial, and wild turkeys obtained from GenBank to evaluate their domestication process, shed light on their origin, trace their worldwide expansion, and define the genetic value of the species, thus emphasizing the repercussions of domestication. Secondly, we developed a haplogroup network by analyzing Nei’s genetic distances and performing an analysis of molecular variance (AMOVA) to assess inter-and intraspecific global connections between turkey populations.

## 2. Materials and Methods

### 2.1. Sample Collection

Blood samples were collected from domestic turkey populations (local breeds) from 2015 to 2018. For each individual, blood samples were taken from the brachial vein, placed in 2 mL vials containing ethylenediaminetetraacetic acid (EDTA), and stored at −18 °C. Additional blood samples were then collected on filter paper sheets and were stored at room temperature until further analysis. Samples were deposited in the Animal Breeding Consulting Laboratory at the University of Córdoba, Spain. A total of 93 samples from local domestic turkey breeds were available for analysis. Eleven samples were collected from rural communities northeast of Paraiba, Brazil; ten samples were from rural communities in Veracruz City, Mexico; ten samples were from Giza Governorate, Egypt; thirty samples were from rural environments in Spain, including eighteen from Andalusia (Sevilla, Cadiz, and Córdoba) and twelve from the Balearic Archipelago (Mallorca); six samples were from a turkey breeding station established in 1983, with birds from northern and northwestern Iran; twenty-two samples belonging to two different breeds were from Italy, with ten samples from the Parma breed and twelve from the Romagnolo breed; and six samples were from Iowa, USA.

### 2.2. Ethics Statement

Ethical approval was not needed for this study. Blood samples were collected from local turkey populations by qualified veterinarians during their routine practice within the framework of official programs aimed at the identification and health monitoring of the breeds and populations included in the present study. Sample collection did not involve any endangered or protected species. The blood samples were manually collected without injuring the animals, and no other types of tissue (e.g., meat) were used in the present study.

### 2.3. DNA Extraction and Amplification

DNA was extracted from paper impregnated with blood samples by using a Chelex-100 resin^®^ (BioRad, Spain) [16]. The mtDNA D-loop sequence was obtained from Genbank accession number EF153719 (*M. gallopavo* mitochondrion 16,717 bp) [17], and Primer3 software (v0.4.0) [18] was used to design primers (F: 5′-CCAAGGATTACGGCTTGAAA-3′ and R: 5′-TCTTCAGTGCCATGCTTTTG-3′) to amplify an mtDNA region of 1248 bp. PCR reactions were performed in a total volume of 25 µL containing 1 × PCR buffer (20 mM of Tris–HCl, pH 8.4; 50 mM of KCl; 1.5 mM of MgCl_2_), 200 mM of each deoxynucleoside triphosphate (dNTP), 10 pmol of each oligonucleotide, 1 U of MyTaq DNA polymerase (Bioline, London, UK), and 30 ng of DNA. The reaction mixtures were placed in a thermocycler (C1000 Touch; Bio-Rad, Hercules, CA, USA) under the following amplification conditions: 95 °C for 5 min; 34 cycles of 95 °C for 45 s, 59 °C for 1 min, and 72 °C for 2 min; and a final extension at 72 °C for 45 min. Amplicon quality was assessed on a 1.5% agarose gel with ethidium bromide, using size (100 bp ladder) and concentration standards. Sufficient amplicon quality was indicated by a single well-defined band of approximately 1000–1300 bp. DNA sequencing was performed by the dideoxy technique on both strands, using a commercial service (Macrogen, Madrid, Spain). In addition to these 93 sequences, we analyzed sequences from wild subspecies (*M. g. merriami*, *M. g. intermedia*, *M. g. osceola*, *M. g. silvestris*, and *M. g. Mexicana*), domestic birds, and commercial lines (improved turkey breeds for meat production) obtained from the NCBI GenBank database (Appendix A), resulting in a total of 542 sequences.

### 2.4. Genetic Diversity and Differentiation

The sequence editing, alignment, and construction of data matrices were performed using Mega v5 [19] and Gblocks 0.91b [20,21]. The number of haplotypes (H), polymorphic sites (S), and nucleotide (π) and haplotype (Hd) diversity estimates for the domestic populations were calculated using DnaSP v5 [21]. An AMOVA [22] was used to calculate genetic variation and differentiation between populations by performing 10,000 permutations. In addition, F_ST_ (permuting haplotypes among population among groups) values from pairwise comparisons were computed with 1000 permutations using ARLEQUIN v3.1 [23].

### 2.5. Genealogical Relationships between Haplotypes

To establish genealogical relationships between haplotypes and their frequencies, a haplotype network was constructed using the median-joining method with the software NETWORK v4.6.0.0 [24] under default parameters. The relationships between haplotypes were also analyzed using phylogenetic inference. Matrices for these analyses included haplotypes that were identified in this study and haplotypes for each subspecies that were reported in the NCBI GenBank database (Appendix A. The models of DNA substitution were evaluated for domestic and wild populations using Splitstree v4.14.6 software [24].

## 3. Results

### 3.1. Sequence Analysis, Genetic Diversity and Differentiation

Ninety-three sequences (620–780 bp long) of the mtDNA D-loop were obtained from the DNA samples of *M. gallopavo* collected for this study. These sequences were registered in GenBank (accession numbers: MK284411-MK284503). Twenty haplotypes were identified within the domestic turkeys, and these all showed overall moderate Hd values and low π values (Table 1). Six haplotypes were identified in domestic turkeys from Mexico. Five haplotypes were detected in birds from Egypt and Iran, with nineteen and eleven polymorphic sites, respectively. These two populations showed high Hd values and moderate π values. In the Spanish populations from Andalusia and Majorca, we detected three haplotypes with five and fourteen polymorphic sites, respectively. Finally, only two haplotypes were identified in the Romagnolo and Parma populations from the Italian and Brazilian populations. Only one polymorphic site was reported for the Romagnolo and Brazilian populations, which also showed moderate Hd values and low π values.

In the domestic local populations used in this study, we detected three haplotypes, designated as MGDH1, MGDH2 and MGDH3. The dominant haplotype was MGDH2 (n = 59), and it was present in 62.76% of the individuals in the population, sharing a maternal line with nine Brazilian individuals, five Egyptian individuals, thirteen Andalusian individuals (Spain), ten Majorcan individuals (Spain), five Mexican individuals, nine Parma individuals (Italy), seven Romagnolo individuals (Italy), and three American individuals (USA). Two domestic individuals from Egypt, four from Andalusia, and three from Iran shared the MGDH1 and MGDH2 haplotypes. Therefore, in the following analyses, they were treated as a single group referred to as “Mundi,” following the premise described by Padilla et al. [1]. Next, to corroborate and strengthen our results, the mtDNA D-loop sequences of additional domestics turkeys (described as *M. gallopavo*) were used to form a group referred to as “Mundi/Domestic.” Sequences of wild individuals reported in the NCBI GenBank database were also included. A summary of the genetic indices for each of the populations studied can be found in Table 2. The total population analyzed (n = 542) showed overall moderate Hd values and low π values (Table 2). The “Mundi/Domestic” turkeys showed an Hd value of 0.562 and a low π value. Among the wild populations, diversity levels varied, with the *M. g mexicana* population showing the lowest Hd and π values. By contrast, the *M. g. osceola* and *M. g. silvestris* populations showed the highest diversity (Table 2).

### 3.2. Analysis of the Internal Genetic Differentiation of the “Mundi” Group

In addition, we analyzed genetic differentiation, including the following external population sequences from Genbank: Oscence turkey samples from Alto Aragon, Spain (OSCE); *M. gallopavo* archaeological (MGA) samples; and samples dating back to 1903, belonging to prehistorical wild *M. g. gallopavo* specimens collected from Veracruz-Llave, Mexico (named 1903). Additionally, we analyzed samples from a commercial line (COMER, meat production turkey,); domestic turkeys from Izabal, Guatemala (GUAT); and domestic turkeys from Michoacan and Puebla, Mexico (DOMEX).

The greatest differentiation in the “Mundi” group was observed between Mexico-Iran, Iran-Brazil, and Romagnolo-Iran turkeys. By contrast, the lowest genetic differentiation values were found between Mexico-Majorca, Parma-Majorca, Parma-Mexico, and USA-Egypt turkeys. The OSCE turkeys showed the highest F_ST_ values of all the populations included in the “Mundi” group. Turkey populations showing the least genetic differentiation were observed between 1903 and Egypt, Andalusia, or the USA; Guatemala and Brazil, Majorca, Mexico, Parma, or the USA; and DOMEX and Andalusia (Spain, Table 3). To corroborate our results, the genetic distances among populations were estimated from mitochondrial sequences using the neighbor-joining method (Figure 1).

We defined two groups of *M. gallopavo*. The first group was divided into the following populations: North America–South America vs. Africa, Europe, and Asia. The second group was divided based on a domestication criterion differentiating America, Europe, and Africa–Asia. The distribution of genetic variation obtained using an AMOVA (*p* < 0.01) for the first defined group (North America–South America vs. Africa, Europe, and Asia) showed that the genetic variation within populations was higher than the genetic variation within groups (Table 4). However, an appreciable amount of variation was also detected within groups (10.40%). The percentage of genetic variation among populations and the low fixation index indicated that subpopulations were not well differentiated and, hence, the populations comprising the species were not structured. When the population was divided into domestication zone groups, the percentage of genetic variation within populations was 83.66%. A low fixation index value, F_CT_, and high variation within populations indicated a lack of genetic structure and differentiation among subpopulations (Table 5).

### 3.3. Haplotype Network

We constructed a two-haplotype network to visualize the relationships between haplotypes and their frequencies. The first network consisted of 294 frequencies and included domestic turkeys from Mexico, commercial lines (industrial turkeys for meat production), archaeological samples, and the “Mundi” group. For the second network, 542 frequencies were used, including the “Mundi” group, wild turkeys, commercial lines, and domestic prehistoric and archeological samples (Appendix A).

The first analysis showed that haplotypes differed from each other by a moderate number of mutations. The network (Appendix A) showed three haplogroups, each one with a dominant haplotype. The biggest haplogroup formed was Meleeagris gallopavo from Mexico (MGFM) n = 108). This group contained eight haplotypes, namely Brazil (nine individuals), Egypt (five individuals), Andalusia (thirteen individuals), Majorca (ten individuals), Mexico (five individuals), Parma (nine individuals), Romagnolo (seven individuals), and the USA (three individuals).

Two domestic individuals from Egypt, four from Andalusia, and three from Iran shared the same haplotype. Most of the domestic turkeys from Brazil, Egypt, Andalusia, Majorca, Italy, Mexico and USA shared the same haplotype and maternal line as the “Mundi” group. This haplogroup comprised Mexican, Guatemalan domestic, and one of the archaeological samples. The second principal haplogroup included eleven “Mundi” haplotypes, which clustered with those obtained from the NCBI GenBank database, comprising 38 individuals related to the dominant haplotype, MGFM (Appendix A).

The third haplogroup detected contained only “Mundi” haplotypes, with the Romagnolo population being most common. The rest of the “Mundi” haplotypes consisted of unique sequences, with one per population from Egypt, Iran, Mallorca, and Mexico; one Parma individual; and two from the USA. Archaeological samples included in this study separated into unique sequences in the network, as seen in Appendix A.

The second analysis revealed 17 haplotypes that differed from each other by a high number of mutations. The network (Figure 2) displayed six haplogroups. H_1 was the main haplogroup and was composed of 318 individuals and seven haplotypes. The dominant haplotypes corresponded to the domestic samples, with most of the “Mundi” and domestic populations included in this haplogroup. In the periphery of these haplogroups, it was possible to observe seven unique sequences beginning to form. Four of these haplotypes belonged to the “Mundi” population. Haplogroup H_2 was located between H_1 and H_3 and contained haplotypes from *M. g. merriani* and archeological and prehistorical domestic samples.

Haplogroup H_3 comprised four haplotypes, all of which consisted of individuals belonging to wild *M. gallopavo*. Dominant haplotypes comprised *M. g. merriani* individuals, who shared a haplogroup with *M. g. silvestris*, archeological samples, and *M. g. intermedia*.

Haplogroup H_4 was shared by prehistorical domestic, *M. g. silvestris*, and *Osceola* haplotypes. The H_5 haplogroup showed eight haplotypes and these were linked to H_1, H_3, H_4, and H_6. The main haplogroup comprised prehistoric domestic haplotypes and was shared by domestic and wild turkey populations of the subspecies *M. g. Mexicana, M. g. osceola, M. g. merriani, M. g. silvestris*, and *M. g. intermedia* and archeological samples. Haplogroup H_6 consisted of four individuals belonging to “Mundi” populations (two from the USA and two from Mexico) and *M. g. silvestris* turkeys. 

## 4. Discussion

Haplotype diversity represents the probability that two randomly sampled alleles are different, while nucleotide diversity is defined as the average number of nucleotide differences per site in pairwise comparisons among DNA sequences, based on the analysis of generated sequences. The results of the present study suggested a moderate level of haplotype diversity and low nucleotide diversity, which may indicate a period of rapid population growth that enhanced the retention of new mutations [25,26].

The results of the analysis of genetic diversity per population revealed that Mexico and USA populations had high Hd and π values. These results differ from those reported for wild turkey populations from the USA, which have high levels of Hd but low levels of π [1,27,28]. Grant and Bowen (1998) suggested these high Hd and π levels may be ascribed to secondary contact between previously differentiated allopatric lineages or to a long evolutionary history in a large stable population; that is to say that the turkey population belonging to the putative place of domestication should have highest mtDNA variability.

The populations from Egypt and Iran showed high values for Hd and low values for π. This condition is often attributed to an expansion process characterized by a period of a low effective population size and a bottleneck, followed by a rapid expansion [1,26]. Similarly, the findings for the Andalusia and Romagnolo populations showed a moderate value for Hd and a low value for π. This has also been reported by Guan et al. [27] in a Spanish black turkey population. These authors ascribed these values for Hd and π as indicative of populations having originated from a small number of founders. The populations from Brazil and Majorca and the Parma population showed low values for Hd and π.

These findings have normally been attributed to different population dynamics, such as periodic bottlenecks in a region or metapopulational structures within regions, directly caused by the low levels of diversity present in the population [26]. The “Mundi/Domestic” turkey population showed high Hd and π values. These results contrast with those obtained by Padilla et al. [1] who reported moderate values for Hd and low values for π in mixed domestic and commercial turkeys.

### 4.1. Haplotype Network

mtDNA diversity patterns often reveal historic migration routes that may date back to the dispersal of the first domesticated animals. Genetic variability is expected to decrease with increasing geographical distance from the epicenter of domestication [29,30] unless an introgression from a wild ancestor species occurred outside the domestication site [30,31]. Qualitatively, the absence of haplogroups in a given region that is still present in neighboring regions indicates a founder effect during or after a gene flow from nearby regions.

Most previous studies dealing with turkey species have only comprised local samples and, hence, have only partially considered haplotype variability. This makes valid conclusions difficult when comparing several international studies. In this context, our study may help clarify such relationships and define turkey mitochondrial haplogroups, since we comprehensively analyzed samples from Mexico and the USA, the countries where the domestication process of turkeys is presumed to have started, but we also included samples from populations in Iran, Egypt, Brazil, Spain, and Italy.

Six divergent groups were identified, but the dominant haplotypes in haplogroup H_1 were the main founding maternal lines of domestic turkeys. On the one hand, since these two maternal lines were detected in individuals from the commercial line and in domestic individuals from Mexico and the USA, these two populations may be the most likely origin of the current highly selected commercial lines. However, the turkey varieties from Andalusia, Majorca, Iran, Egypt, and Italy and the commercial populations may have derived from the wild Mexican population. This suggests that commercial lines may share the same origin and that specific nuclear DNA haplotypes may have been indirectly selected in modern domestic turkey populations.

### 4.2. Analysis of the Internal Genetic Differentiation of the “Mundi” Group

While population genetic diversity data suggested that Mexico and USA populations were consistently more diverse than the other populations analyzed, in the case of the USA, the high level of divergence between haplotypes may be attributed to secondary contact between previously differentiated allopatric lineages and the stable population settled in the area.

This may have occurred when British Pilgrims brought domestic European turkeys back to eastern North America in the 17th century. These reintroduced birds hybridized with the American eastern wild turkey (*M. g. silvestris*), resulting in the ancestors of the modern commercial turkeys [9].

In Mexico, the high level of diversity may be attributed to the fact that the local population from this area is a large stable population with a long evolutionary history. Domestic Mexican turkeys tended to be among the most diverse populations, while Brazilian turkeys tended to be the least diverse, particularly with respect to mtDNA diversity values. The remaining domesticated populations showed a general pattern of moderate haplotype diversity and low nucleotide diversity among mitotypes.

Grant and Bowen [26] suggested that such a pattern could be ascribed to a recent expansion from a period in which there was a low population size and where low sequence diversity among mitotypes may have been attributed to a recent coalescence event. In this context, the high nucleotide diversity may be a result of the retention of mutations in an expanding population.

The F_ST_ pairwise values estimated for the nine “Mundi” populations are reported in Table 4. After analyzing the distances between the populations studied, it became evident that Parma and Majorca populations showed no significant genetic differences or variation with respect to the Mexican population. The Egyptian population did not differ from the USA population, which was the population with the second highest nucleotide and haplotype diversity. It should be noted that negative F_ST_ values should be effectively seen as zero values. A zero value for F_ST_ indicates that there is no genetic subdivision between some populations that were included in the study.

By contrast, when Mexican and USA populations were compared, significant differences in diversity were observed. After comparing them to populations outside the haplogroup (OSCE, MGA, 1903, COMER, GUAT, and DOMEX), we can infer that “Mundi” populations had greater variability and were more distant from the COMER, MGA, and OSCE populations. Archaeological samples from populations dating back to 1903 [5] and GUAT and DOMEX populations were shown to differ in terms of their variability and showed a low-to-moderate distance. The population from Iran was the most genetically distant from the rest of analyzed populations, being very close to the COMER populations. This information was inferred from the neighbor-joining tree distance (Figure 1). By analyzing pairwise genetic differentiation and the distribution of genetic variation through the AMOVA analysis, it was possible to observe genetic differences within populations. These results indicated that, although these groups shared common haplotypes and a common maternal line, mtDNA variation in the populations and the proportion of unshared haplotypes in each group was significant and variable [1,32].

## 5. Conclusions

Our results showed a moderate haplotype diversity and a low nucleotide diversity, which may be attributed to a period of population expansion followed by a period of low effective population. The majority of domestic turkey breeds in Brazil, Egypt, Andalucia, Mallorca, Italy, Mexico, and the USA were seen to share the same haplotype and maternal line. Our results suggest that turkey populations from Mexico and the USA were historically large and stable populations that occupied a wide geographical distribution ranging from the north to the center of the American territories. These populations then thrived and became stable and more diverse than the other populations over the years. Such success may be attributed to the conservative management practices used in rural areas that did not permit the introduction of commercial lineages or improved crossbred turkeys for production. Turkey populations from rural communities in Andalusia, Majorca, Egypt, Iran, Italy, and the USA consisted of six haplotypes, which has not previously been reported for *M. gallopavo* populations.

## Figures and Tables

**Figure 1 animals-09-00897-f001:**
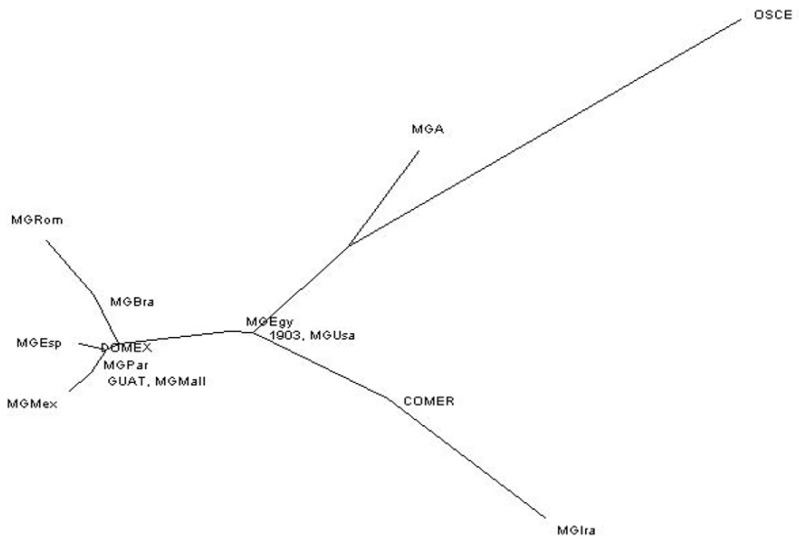
Neighbor-joining distance tree obtained after 5000 bootstrap replicates. MGBra: Brazil, MGEgy: Egypt, MGAnd: Andalusia, MGIra: Iran, MGMall: Majorca, MGMex: Mexico, MGPar: Parma, MGRom: Romagnolo, MGUsa: USA, OSCE: Oscence, MGA: *Meleagris gallopavo* archeological, 1903: *Meleagris gallopavo* archeological dating back to 1903, COMER: commercial breeds, GUAT: domestic breeds from Guatemala, DOMEX: domestic breeds from Mexico.

**Figure 2 animals-09-00897-f002:**
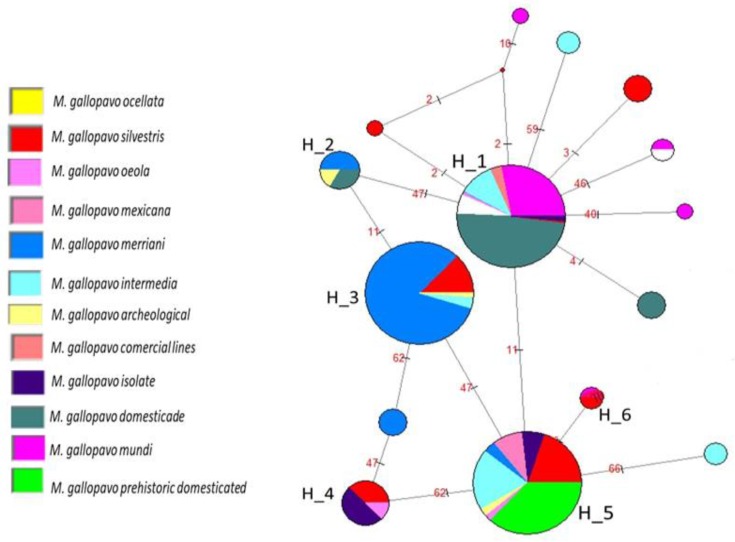
Median-joining haplotype network. Relationships are depicted between haplotypes identified in domestic and wild *M. gallopavo*, including 542 individuals and 17 haplotypes.

**Table 1 animals-09-00897-t001:** Genetic diversity indices for each population in the study.

Population	N	Tnm	H	Hd	π	S
Egypt	10	19	5	0.844	0.0070	19
Brazil	11	1	2	0.327	0.0005	1
Andalusia (Spain)	18	5	3	0.451	0.0013	5
Iran	6	11	5	0.933	0.0058	11
Majorca (Spain)	12	14	3	0.318	0.0037	14
Mexico	9	40	6	0.833	0.0175	40
Parma (Italy)	10	4	2	0.200	0.0012	4
Romagnolo (Italy)	12	1	2	0.530	0.0008	1
USA	5	16	3	0.700	0.0102	16
Total	93	79	20	0.596	0.0047	76

N: number of individuals; Tnm: total number of mutations; H: number of haplotypes; Hd: haplotype diversity; π: nucleotide diversity; S: number of polymorphic sites.

**Table 2 animals-09-00897-t002:** Genetic diversity indices for each population studied.

Population	N	H	S	Hd	π	Tajima’s D Value
*Mundi/Domestic*	273	21	37	0.562	0.00215	−2.38448 (NS, *p* < 0.01)
*M. g. merriami*	86	14	13	0.724	0.00441	0.69964 (NS, *p* < 0.10)
*M. g. intermedia*	77	13	13	0.887	0.00574	−0.15106 (NS, *p* < 0.10)
*M. g. osceola*	8	8	8	1	0.00538	−1.14142 (NS, *p* > 0.10)
*M. g. silvestris*	77	22	23	0.923	0.00516	−1.57292 (NS, 0.10 > *p* > 0.05)
*M. g. mexicana*	36	6	9	0.352	0.00221	−1.65410 (NS, 0.10 > *p* > 0.05)
*Total*	542	17	56	0.643	0.02442	−2.24320 (NS, *p* < 0.01)

N: number of individuals, H: number of haplotypes, S: polymorphic sites, Hd: haplotype diversity, π: nucleotide diversity, NS: Not significant.

**Table 3 animals-09-00897-t003:** Pairwise genetic differentiation (F_ST—_permuting haplotypes among population among groups) of populations.

	MGBra	MGEgy	MGAnd	MGIra	MGMall	MGMex	MGPar	MGRom	MGUsa	OSCE	MGA	1903	COMER	GUAT
MGBra	0													
MGEgy	0.041	0												
MGAnd	0.138	0.104	0											
MGIra	0.755	0.148	0.623	0										
MGMall	0.032	0.076	0.082	0.636	0									
MGMex	0.057	0.074	0.098	0.791	−0.013	0								
MGPar	0.040	0.080	0.090	0.662	−0.000	−0.005	0							
MGRom	0.038	0.022	0.309	0.709	0.202	0.268	0.213	0						
MGUsa	0.139	−0.000	0.153	0.288	0.105	0.105	0.090	0.203	0					
OSCE	0.907	0.385	0.853	0.868	0.821	0.940	0.846	0.861	0.570	0				
MGA	0.469	0.321	0.544	0.495	0.455	0.442	0.436	0.485	0.257	0.608	0			
1903	0.302	−0.258	−0.093	0.225	0.097	0.352	0.120	0.311	−0.275	0.893	0.218	0		
COMER	0.443	0.149	0.343	0.020	0.404	0.427	0.399	0.466	0.185	0.511	0.489	−0.163	0	
GUAT	−0.040	0.043	−0.005	0.825	−0.128	−0.116	−0.121	0.200	−0.052	1.000	0.320	0.384	0.337	0
DOMEX	0.103	0.250	−0.023	0.458	0.095	0.067	0.092	0.253	0.219	0.711	0.702	−0.180	0.376	−0.018

MGBra: Brazil, MGEgy: Egypt, MGAnd: Andalusia, MGIra: Iran, MGMall: Majorca, MGMex: Mexico, MGPar: Parma, MGRom: Romagnolo, MGUsa: USA, OSCE: Oscence, MGA: *Meleagris gallopavo* archeological, 1903: *Meleagris gallopavo* archeological dating back to 1903, COMER: commercial breeds, GUAT: domestic breeds from Guatemala, DOMEX: domestic breeds from Mexico.

**Table 4 animals-09-00897-t004:** Summary of analysis of molecular variance (AMOVA) significant value results (*p* < 0.01) without a priori assumptions, defined by population.

Source of Variation	Df	Sum of Squares	Variance Components	Percentage of Variation	Fixation Indices
Among groups	1	32.51	0.12069 Va	7.03	
Among populations/ within groups	3	14.207	0.17839 Vb	10.40	F_SC_: 0.11182 F_ST_: 0.17429 F_CT_: 0.07033
Within populations	538	762.335	1.41698 Vc	82.57	
Total	542	811.794	1.71607		

Group: North America-South America vs. Africa-Europe-Asia. Variance for group, Va; variance for population, Vb; variance for haplotypes within a population within a group, Vc; permuting haplotypes among population among groups, F_ST_; permuting haplotypes among populations within groups, F_SC_; permuting populations among groups, F_CT_.

**Table 5 animals-09-00897-t005:** Summary of AMOVA significant value results (*p* < 0.01) without a priori assumptions, defined by domestication zone.

Source of Variation	Df	Sum of Squares	Variance Components	Percentage of Variation	Fixation Indices
Among groups	2	38.284	0.05577 Va	3.26	
Among populations/within groups	1	3.104	0.22331 Vb	13.07	F_SC_: 0.13513 F_ST_: 0.16336 F_CT_: 0.03265
Within Populations	539	770.405	1.42932 Vc	83.66	
Total	542	811.794	1.70841		

Groups: “America vs. Africa-Asia vs. Europe.” Variance for group, Va; variance for population, Vb; variance for haplotypes within a population within a group, Vc; permuting haplotypes among population among groups, F_ST_; permuting haplotypes among populations within groups, F_SC_; permuting populations among groups, F_CT_.

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
