# Peer review of "Tracing Worldwide Turkey Genetic Diversity Using D-loop Sequence Mitochondrial DNA Analysis"

_animals, 2019, doi:10.3390/ani9110897_

Round 1

Reviewer 1 Report

The ms is plagued by a large number of weaknesses, flaws, errors, etc. The abstract and introduction have many genetic sentences that are difficult to understand or guess what they mean or imply. For instance, what do you mean by "their roots", "internal genetic background"? what kind of "model" are you referring to? "international turkey populations"? "Standard mitochondrial DNA analysis"? Please be clear. 

MtDNA based Molecular studies have addressed domesticated turkey to be representative of the extinct wild subspecies (M. g. gallopavo) [5]. Simultaneously, historical registries address different prehispanic Mexican groups like Teotihuacan ones, among other domesticated ethnic groups.....? I can't understand this. 

I was quite confused in some parts of the ms. Which samples are from domesticated animals? which ones from commercial? and which ones from wild turkeys? what is the actual difference between domesticated and commercial animals?

To corroborate our results, we designed an NJ (Neighbor-joining) distance (Figure 1)? We designed? distance of what? 

Why did you get negative Fst values?

Title: turkey biodiversity? 

Author Response

Response to Reviewer 1,

The ms is plagued by a large number of weaknesses, flaws, errors, etc. The abstract and introduction have many genetic sentences that are difficult to understand or guess what they mean or imply. For instance, what do you mean by "their roots", "internal genetic background"? what kind of "model" are you referring to? "international turkey populations"? "Standard mitochondrial DNA analysis"? Please be clear. 

Response: For a better understanding of ms these suggestions have been modified in the text, line 24, 29, 35 and 41. Besides a complete English and syntax revision have been made by an expert.

mtDNA based Molecular studies have addressed domesticated turkey to be representative of the extinct wild subspecies (M. g. gallopavo) [5]. Simultaneously, historical registries address different prehispanic Mexican groups like Teotihuacan ones, among other domesticated ethnic groups......? I can't understand this. 

Response:  This paragraph was modified for a better understanding:

Line 69 to 74: MtDNA based Molecular studies have been performed on domestic turkey population in order to trace the history of extinct wild subspecies (M. g. gallopavo) [5]. Historical registries suggest that, different prehispanic Mexican groups like Teotihuacans, began the domestication of turkey for the first time already in the period ranging from 200 and 700 BC [4]

I was quite confused in some parts of the ms. 

Which samples are from domesticated animals?  

Response: The 93 samples of our study are from domestic local turkeys , that is possible to find in houses backyard, ensuring that they did not have contamination with the commercial line, and we take Genbank sequences of other domestic turkey from Mexico and we call this group Mundi/domestic

which ones from commercial?  

Response: we clarified this point in the manuscript

 and which ones from wild turkeys?

Response: We clarified this point on the line 152, Meleagris gallopavo  merriami, Meleagris gallopavo intermedia, Meleagris gallopavo osceola,  Meleagris gallopavo silvestris and Meleagris gallopavo  mexicana

what is the actual difference between domesticated and commercial animals? 

Response: we clarified this difference in the manuscript, but in the current study we refer to as commercial all those bird’s domestic turkey of commercial genetic lines that are used for meat worldwide production and we refer to domesticated to all local breeds populations can be raised in more arid or hot conditions than in the natural distribution sites; because they tolerate heat better and are adapted to shepherd, we change the word domesticated for domestic or domestics in some paragraphs

To corroborate our results, we designed an NJ (Neighbor-joining) distance (Figure 1)? We designed? distance of what?

Response: This sentence was modified as follow,

Line 224: To corroborate our results, genetic distances among populations, namely NJ (Neighbor-joining) was estimated from mitochondrial sequences (Figure 1).

Why did you get negative Fst values?

Response: Line 389 In the manuscript was described. A negative value means that no differences were found between the populations, the negative value is equal to 0. This is probably due the short time from domestication that did’t allow a strong differentiation in dloop region or to a recent admixture event

Reviewer 2 Report

The authors answered comments raised in previous review.

Author Response

Response to reviewer 2

The manuscript have been submitted to a English editing services

Reviewer 3 Report

Overall, this manuscript is well-written and useful for the turkey society. The authors studied the phylogenetic and genealogical basis of the current turkey population using mitochondrial DNA that were collected from 93 turkey samples worldwide. Their results could be useful for understanding the biodiversity of turkey around the world, and for designing the conservation strategies, as well as for the breeding industry, when integrating them with whole genome sequencing data later if possible. The methods they used are standard and the statistics behind the results are solid. I have few major comments on the manuscript. The English writing may need double check in some places in the manuscript. For instance,

1). Line 33, could be "the phylogenetic and genealogical background underlying internal turkey populations are still..."

2). Line 76, how about replacing "used" with "explored"

Author Response

Response to Reviewer 2,

1). Line 33, could be "the phylogenetic and genealogical background underlying internal turkey populations are still..."

Response: Line 33 was modified and now is the line 34, “the phylogenetic and genealogical background underlying internal turkey populations are still"

2). Line 76, how about replacing "used" with "explored"

Response: The word used was replacing with explored, line 83

Round 2

Reviewer 1 Report

The ms has improved and I only have minor comments:

Title: change biodiversity by genetic diversity

Lines 39-42: Polymorphic sites and genetic diversity are related. So, saying that Mexico has the greatest number of polymorphic sites and the highest genetic diversity is the same. Please re-write.

Please improve the quality of the Figures

Line 303: remove "our" 

Author Response

Response to Reviewer 1

The ms has improved and I only have minor comments:

Title: change biodiversity by genetic diversity

R: The title was change for genetic diversity

Lines 39-42: Polymorphic sites and genetic diversity are related. So, saying that Mexico has the greatest number of polymorphic sites and the highest genetic diversity is the same. Please re-write.

R: The Lines were the lines were modified,

Turkeys from Mexico showed the greatest number of polymorphic sites (40), while turkeys from Italy and Brazil reported only one site each. Nucleotide diversity was also highest in Mexico and the USA (π = 0.0175 and 0.0102, respectively) and lowest in Brazil and Italy.

Please improve the quality of the Figures

Line 303: remove "our"

R: The word our was removed

This manuscript is a resubmission of an earlier submission. The following is a list of the peer review reports and author responses from that submission.

Round 1

Reviewer 1 Report

This study analyzes biodiversity of turkey, using samples from domesticated animals from several countries. This might be one of the first studies that analyzes relatively comprehensively turkeys not just from the place of its domestication (Mexico, North America), but from worldwide locations. Standard mitochondrial DNA analysis (D-loop sequence) was used, and analysis of genetic diversity and haplotype network was performed on the samples (six major haplogroups were defined), including publicly available sequences from previous studies.

The aim of the work and the scope of the analyzed samples is of high interest for research regarding this important domesticated bird. However, the manuscript in the current state has major deficiencies. It contains very high amount of language and spelling errors and all kinds of typos. The flow of the text is quite chaotic and very hard to follow. Results should be described clearly, with short summary for every results section. Discussion again is hard to follow, with very vague sentences of this type:

“However, most of the previous studies were based on local samples and thus only considered a part of the whole haplotype variability, what makes difficult to make the correspondence between several studies analysing samples from different geographic origins.”

Also, better description of samples is necessary (e.g. not just „six from Iran“ or „ten from Egypt“.

Reviewer 2 Report

International assessment of Turkey biodiversity using mitochondrial DNA by Canales and collaborators

This study examines mtDNA to “investigate the genetic diversity and structure, genetic relationships and the phylogenetic relationships” of domestic, commercial and wild populations of Turkey (Meleagris gallopavo).

I found this manuscript hard to follow. It needs to improve its organisation and style. There are many flawed statements, some methodological weaknesses and the English need to be improved before it will consider for publication. I recommend rejecting this manuscript.

What is the sequence divergence between the haplogroups? It is a mistake to give a “nomenclature” when it was not evaluated and when those haplogroups shared haplotypes and/or are only separated by a few mutations. Also, you need to consider biological processes involved in genetic variants such as population size or the nature of the molecule used (i.e. high mutation rates and no recombination). Also, according to the AMOVA, most variation is found within populations rather than among populations.

There are several terms that have been used incorrectly. For instance, mutations, polymorphic sites or segregating sites are commonly used indistinctively. So, in lines 140-142 you are saying the same that in lines 150-151 although the values are not the same for all populations (73 in the first part and 76 in the second part) but still the same for Mexico.

It is difficult to follow when you used indiscriminately country names or city/place names. I suggest that you stick only to country names.

What are you referring to “models of distance” in line 133? Are you probably referring to models of DNA substitution?

I think the total number of samples is not 93. 11(Brazil)+10(Mexico)+10(Egypt)+30(Spain)+6(Iran)+32(Italy)+6(USA) =105

Or is the total from Spain 18 samples?

Lines 55-56: Knowledge from historical registries indicate that different prehispanic Mexican groups like Teotihuacans and other ethnic groups of domesticated turkeys….???

Line 69: South of America or South America?

The methods section needs a better organisation. There are several new things in the results that should go to the methods. Also, where is the data from line 183 onwards coming from?

Lines 62-65: Too many ideas in a single sentence.

The title needs to be changed. I suggest not to use “International assessment”. Please give a more meaningful title.